# XploreML: An interactive approach to study and explore machine learning models

## Abstract

Due to its achievements in recent years, Machine Learning (ML) is now used in a wide variety of problem domains. Educating ML has hence become an important factor to enable novel applications. To address this challenge, this paper introduces XploreML – an interactive approach for lecturers to teach and for students or practitioners to study and explore the fundamentals of machine learning. XploreML allows users to experiment with data preparation, data transformation and a wide range of classifiers. The data sets can be visually investigated in order to understand the complexity of the classification problem. The selected classifier can either be autonomously fitted to the training data or the effect of manually altering model hyperparameters can be explored. Additionally source code of configured ML pipelines can be extracted. XploreML can be used within a lecture as an interactive demo or by students in a lab session. Both scenarios were evaluated with a user survey, where both variants were assessed as positive with the first yielding more positive feedback. XploreML can be used online: `ml-and-vis.shinyapps.io/XploreML`

## 1. Introduction

XploreML is an interactive web-based tool running inside a webbrowser. Target users are lecturers aswell as students and practitioners learning the fundamentals of machine learning. XploreML enables the users to study supervised machine learning (ML) models and explore the effects of hyperparameters and data preparation. It is best suited for beginner's courses or self-study to accompany the introduction of ML methods, in specific classification. It can be used (a) by a lecturer enhancing a theoretical lecture with an interactive demo or (b) by students in a lab session. Both scenarios were evaluated with a user survey.

Provided or own data sets can be visually analysed, scaling and projection methods can be applied, and a variety of classifiers can be trained on the data. The classifiers are selected and trained with the configured ML pipeline or the user can interactivelly experiment with hyperparameters. With a hands-on-approach users are encouraged to:

1. understand how different models classify data with different class separability

2. explore the effect of model hyperparameters

3. investigate the effects of scaling and transforming the input data

4. develop an intuition of possible results for different data sets and classifiers

5. learn programming of ML by extracting source code snippets of configured ML pipelines

## 2. Related work

Similar approaches have been proposed. An approach to educate beginners in ML was proposed in (Beitner et al., 2018), with one data set and four classifiers. In contrast, XploreML offers a variety of data sets and classifiers, but assumes some basic knowledge of ML. In (Weidele et al., 2020) an advanced approach visualizing hyperparameter tuning is proposed, but not with an educational focus. Tensorflow playground (tfp) allows for detailed exploration of hyperparameters but is constrained to neural networks. The groundbreaking tool Weka (Witten & Frank, 2005) has a huge variety of ML models, but requires setup. Subsequent to using XploreML, Weka would be a good option to further refine and widen the gained knowledge.

## 3. System description XploreML

XploreML is a web-based tool implemented in R (R Core Team, 2017) with caret (Kuhn, 2017), lattice (Sarkar, 2008), and shiny (Chang et al., 2017). It can be used online[1] with-

---

[1]Anonymous Institution, Anonymous City, Anonymous Region, Anonymous Country. Correspondence to: Anonymous Author <anon.email@domain.com>.

Preliminary work. Under review by the International Conference on Machine Learning (ICML). Do not distribute.

---

[1]XploreML: `ml-and-vis.shinyapps.io/XploreML`

| classifier | hyperparameter |
|---|---|
| linear discr. analysis | - |
| quadratic discr. analysis | - |
| k-NN (Han et al., 2012) | $k$: number of neighbors |
| SVM (linear soft-margin) (Abe, 2010) | $C$: regularization for slack var. |
| SVM (RBF-kernel) (Abe, 2010) | $C$: regularization for slack var. $\sigma$: standard deviation of kernel |
| SVM (polynom. kernel) (Abe, 2010) | $C$: regularization for slack var. *degree*: degree of polynomial |
| decision tree (CART) (Breiman et al., 1984) | *maxdepth*: max. depth of tree |
| random forest (Breiman, 2001) | *mtry*: number of features at split |
| xgboost (Chen & Guestrin, 2016) | *maxdepth*: max. depth of tree $\eta$: learning rate $\gamma$: minimum loss reduction |
| artificial neural network (Han et al., 2012) | *size*: nodes in hidden layer *decay*: weight decay |
| OneRule and rule base | - |

*Table 1.* Classifiers available in XploreML with their used hyperparameters. These can be manually adjusted in the *Experimentation* pane and are used during training in hyperparameter tuning in the *Classification* pane.

out setup for provisioned and own data sets of small scale and a video is available[2].

The user interface is subdivided into a workflow configuration pane on the left and the output pane on the right (see Fig. 1). The ML pipeline is configured with the following steps:

- *Data acquisition:* Different data sets can be selected to explore pros and cons of different models and data preparation steps – from an easily separable artificial 2 clusters data set to real educational data sets like iris or wine (Dua & Graff, 2020). In addition own small-scale data sets can be uploaded.

- *Pre-processing:* In order to understand the effect of scaling the input data, z-score, min/max-scaling, or no scaling can be applied.

- *Transformation:* The data can be projected onto two dimensions (Sacha et al., 2017) with PCA, ICA, and t-SNE (van der Maaten & Hinton, 2008).

- *Model selection:* Users can select from a range of classifiers (see Table 1). The available classifiers were carefully selected to allow for experimentation with different families of classifiers. In addition the sampling method can be selected.

---

[2] XploreML video: `youtu.be/FzKfHLt27cw`

Each change in the settings re-runs the workflow and the results and visualizations are shown in the output pane on the right (see Fig. 1) consisting of following subpanes:

- *Getting started:* To get started the first steps are described, a video is embedded and a brief user manual together with links to web resources introducing the methods is offered.

- *Visualization:* A data set's value ranges, distributions and class overlaps are shown with box or density plots. Correlations are investigated with a scatter plot matrix or parallel coordinates. This allows to evaluate class separability, understand the complexity of the classification problem, and the necessity of scaling can be evaluated. In addition feature correlations and feature importance is shown.

- *Classification:* The classifier is fitted to the train set by tuning the hyperparameters using cross validation or bootstrap (Han et al., 2012). The best model is then used for classification and its hyperparameters, the classified data, and classification metrics are shown. In order to get familiar with programming of ML pipelines, the R source code snippet of the currently configured pipeline can be viewed and downloaded.

- *Experimentation:* The effect of hyperparameters can be intuitively explored in a 2D-scatter plot. For $> 2$ attributes the data can be projected or the first two attributes are used. The hyperparameters (see Table 1) can be interactively altered, which re-trains the model and updates the decision function (see Fig. 2). The different models' decision functions can be evaluated allowing to compare linear, piecewise-linear and fully flexible decision functions. This can be used to experiment with underfitting vs. overfitting.

The design of XploreML is in several ways in accordance with the curriculum development handbook in (Becker & Michonneau): A backward design was used, i.e. the desired outcome was first defined and the tool was tailored towards the lecture and lab sessions accordingly. In order to support learning, "scaffolding" is offered by allowing to extract source code of ML pipelines. In addition, the suggestion of using a single data set to introduce several ML topics was used in the lab session (see Section 4).

In the following, two examples of using XploreML when teaching or learning ML fundamentals are briefly described:

1. The necessity to scale input data can be shown by using the scale-sensitive k-NN classifier on provisioned unscaled data (e.g. XOR_streched). Though the two classes should be easily separable, the attributes' highly

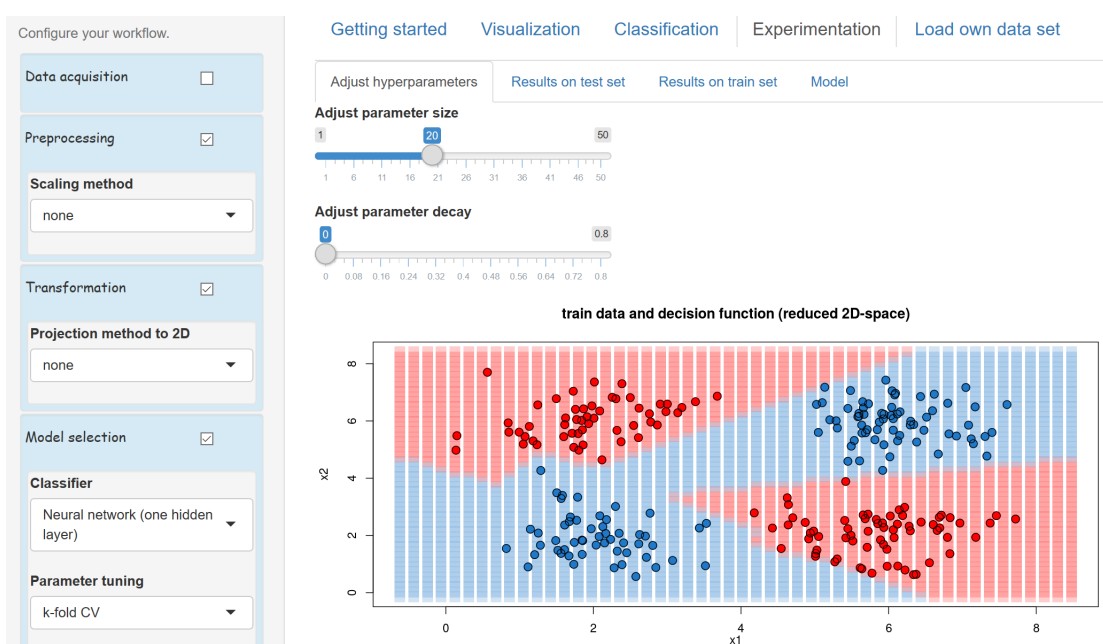

*Figure 1.* XploreML: In the left pane, the ML pipeline is configured by selecting data sets, attributes, scaling and projection methods and classifiers. The right pane shows visualizations and metrics of the input data (*Visualization*), the classification results (*Classification*), lets the user experiment with hyperparameters (*Experimentation*), allows to upload own data sets and to view and download the extracted executable R source code snippet of the currently configured pipeline.

different value ranges prevent the distance-based k-NN classifier from finding a good decision function.

2. Fig. 2 shows experimentation with hyperparameters of an RBF-SVM on the two-dimensional PCA-projection of the 13-dimensional wine data set (Dua & Graff, 2020). Keeping the regularization parameter $C$ at low values ($C = 0.2$), $\sigma = 12$ yields a poor accuracy of $0.771$. Adjusting $\sigma$ to lower values ($\sigma = 0.8$) increases the accuracy on the test set to $0.957$.

## 4. Using XploreML to teach machine learning

XploreML lends itself to be used within a lecture as an interactive demo with questions and discussions accompanying the teaching of ML fundamentals.

Alternatively it can be used by students as part of a lab session. In a lab session the following tasks were given out to students of a beginner's course in machine learning. The underlying data set was the 13-dimensional red wine data set with three classes (Dua & Graff, 2020):

1. *"Analysis of input data: Using the* Visualization *pane, find out: (a) characteristics of the data set (classes, features, value ranges, etc.), (b) how well the data is separable, and (c) which are good features in order to separate the classes."*

2. *"Effects of scaling: Using the* Classification *pane, classify the data set with k-NN. Why is the result so poor? Use scaling methods and classify again. Now use decision tree and random forest with and without scaling. What are your findings with/without scaling?"*

3. *"Projection methods: Transform the data set to 2 dimensions. Classify the data set using classifiers of your choice. What are your findings regarding accuracy and elapsed time for training?"*

4. *"Effects of hyperparameters: Using the* Experimentation *pane, experiment with hyperparameters in order to understand their effects. Use the projected data set and the classifiers decision tree, SVM (polynomial, softmargin), and neural network. What are your findings for each of the classifiers?"*

5. *"Best ML pipeline: Find the best ML pipeline for the data set. Hyperparameter tuning takes place automatically, so the task for you is to find the best combination of feature selection, preprocessing, transformation, and classifier. What are your results and what is your selected ML pipeline?"*

## 5. Evaluation: user survey

The scenarios of (a) using XploreML as a lecturer accompanying the lecture and (b) using it in a lab session were

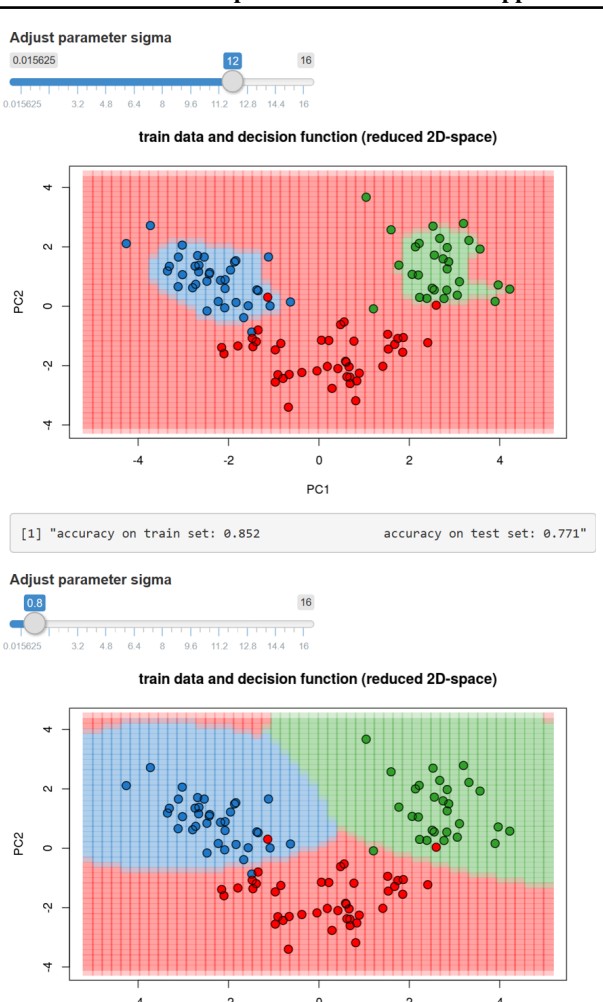

*Figure 2.* Experimenting with hyperparameters of an RBF-SVM on the PCA-projected 13-dimensional wine data set (Dua & Graff, 2020).

evaluated in a user survey. The tool was used after preprocessing, transformation and traditional machine learning methods were covered in the course.

The evaluation was conducted with different groups of students, where for group A (N=16 participants) XploreML was integrated into the lecture and for group B (N=21 participants) it was part of a lab session with the students using the tool. Student feedback was acquired with a brief anonymous questionnaire with three questions Q1...Q3, where the answer options correspond to a Likert scale. The questions and summarized results are shown in Table 2 and the detailed results are depicted in Fig. 3 with diverging stacked bar charts (Heiberger & Robbins, 2014).

While the approach was predominantly assessed as posi-

| question | scenario | mode | mean |
|---|---|---|---|
| Q1: Was the tool helpful to understand scaling of input data? | lecture lab | definitely yes yes | 4.62 3.62 |
| Q2: Was the tool helpful to understand the effects of hyperparameters? | lecture lab | definitely yes neutral | 4.5 3.19 |
| Q3: Would you use the tool for your studies? | lecture lab | definitely yes yes | 4.75 3.67 |

*Table 2.* Questions and results of user survey, where due to the ordinal Likert scale the mode (most frequent answer) is reported and the mean value is given for completeness. Answer options were [1] definitely no ... [5] definitely yes.

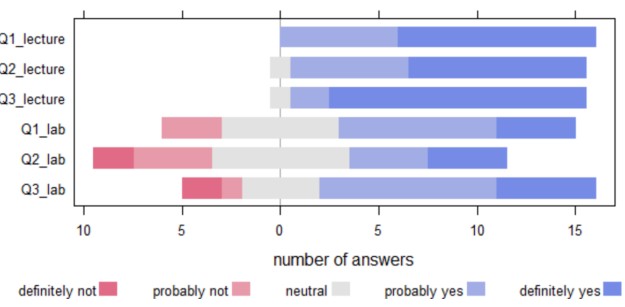

*Figure 3.* Results of user survey showing the number of answers per questions in each of the categories. (Note that for Q1_lecture - Q3_lecture N=16 and for Q1_lab - Q3_lab N=21 participants.)

tive, it is an interesting finding, that the scenario where the approach is used by the lecturer to accompany teaching were viewed as more positive. As a consequence of this finding, the feature to extract source code snippets from configured ML pipelines was added. This links the use of XploreML more closely with exercises or projects requiring programming of own ML pipelines.

## 6. Conclusion

It was shown how the introduced tool XploreML can support the teaching and learning of ML fundamentals. The scenarios of the lecturer using it to accompany the introduction of ML basics and the use of the tool by students in lab sessions were comparatively evaluated in a user study with positive assessment. This indicates that XploreML lends itself to be used by both, lecturers and students. XploreML is used in the author's lectures and lab sessions and is available online without setup. Lecturers and students are invited to test it.

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
