# OpenReview forum: "XploreML: An interactive approach to study and explore machine learning models"
_ECMLPKDD.org/2020/Workshop/TeachML — Submitted to ECML PKDD 2020 TeachML_

### Official Review · AnonReviewer1 · 2020-07-16
**Great potential for experimenting with different models, but implementation lacks quality**

**Rating:** 6
**Confidence:** 4

**Review:**

This *could* be a great resource for experimentation in supervised learning without any programming. The app is well-documented and covers most classifiers that a standard intro to supervised ML would contain.

For courses that are based on the software stack used here (R + caret), the fact that the code generating the models can be inspected, too, is even more valuable.

Unfortunately, the implementation seems to be very unstable, buggy and slow:

- xgboost did not work on any of the datasets I tried, it would hang for a couple of minutes and then spit out a generic error message
- results from previous runs were not updated, so one tab would show classification boundaries for a decision tree, say, while the other tab would still show confusion matrices for a LDA model computed earlier. Didactically, that's terrible, because students will not have the self-confidence and experience to recognize the wrong/incongruopus results presented by the app. At the very least, the interface should be programmed so that results that are "out of date" are greyed out until they are recomputed for the current model.
- the server hosting it was very unresponsive for most of the time I tried to experiment with it, loading the app and reccomputing even very simple models like LDA took a (very) long time on all three occasions that I used it
- there seems to be no way to abort computations once they are under way and users lose patience. This is not a good UX.

In terms of functionality, I would have liked to see the possibilty of tuning/changing more than 1 hyperparameter for some of the methods, sometimes the interplay of them is very important (e.g. max tree depth and min node size), and having only one of the many hyperparameters configurable is likely to lead to misunderstandings like "this is the only tuning parameter that matters" on the side of the students.

Also:
Paper needs a spell check ("aswell", "interactivelly" etc) and should be proof-read by a dilligent native speaker ("in specific classification" -> "specifically classification", lots of other weird phrases).

Also:
Code for the Shiny app should be made public to enable local hosting instead of relying on RStudio's rather limited free hosting.

**Summary:**
This is a fairly ambitious project with large potential, not all that well executed. I don't think I would use it in my teaching in its current state, too many frustrating bugs and inconsistencies and too much waiting around on the unresponsive server (although the shiny app could be easily hosted locally if the authors make their code public, see below, and this would presumably speed up computation and rendering quite a bit)

---

### Official Review · AnonReviewer3 · 2020-07-28
**Interactive approach for learning ML Classification models**

**Rating:** 5
**Confidence:** 5

**Review:**

XploreML presents an interactive GUI for learning fundamentals of supervised classification models. This app has potential to be used as a training tool, however, implementation lacks clarity and stability.

- One major issue is that sever gets disconnected quite often and it is set to default settings when reloaded again. This means that while mentors/students are discussing aspects of current implementation the results will be gone and everything needs to be set again from scratch.
- The app offers selection of only a few hyperparameters. Learning rate in the neural net, depth in random forest are important adjustable hyperparameters. Also, the number of CV folds and bootstraps and train, validation and test split should be adjustable.
- The raw data tab in visualization pane does not show class labels. Complete visualization of raw data with class labels is a very fundamental aspect of beginning ML teaching/learning process.
- Furthermore, in the visualization of data the ‘parallel coordinates’ does not help understand data for beginners rather it looks confusing. For beginners, clear plots should be used such that students can easily read and understand data stats themselves without a mentor’s intervention. Plots of data with more than 4 features are almost unreadable.
- While running a new model, the results and stats from the previous run remain intact, which is very confusing.
- A very basic model taught in supervised binary classification is the logistic regression model which is missing in this app. Although LDA is closely related to logistic regression, but for beginners, it will make more sense to make them familiar with logistic regression for classification than LDA.

Summary:
The authors propose an interactive GUI for exploring basic supervised classification models, however, it is not very handy in its current state. It offers only a few adjustable parameters, inconsistent view, and unstable server.  All these factors make it unlikely to be accepted.

---

### Official Review · AnonReviewer2 · 2020-07-28
**Interesting tool, but unclear how it fits into a broader ML learning experience**

**Rating:** 4
**Confidence:** 5

**Review:**

XploreML is an interactive tool that allows for students to explore various classification algorithms dynamically. The paper's current presentation of XploreML leaves it to the reader to determine the learning goals of this tool. Additionally, while the authors discuss using backward design in their implementation for using this tool, without the learning goals specified, it is challenging to understand where and how this tool comes into a machine learning course. Is it the first, second, last, etc contact for a student learning a concept? Finally, one of the features that I was most interested in was the code extraction. I felt that this was shortchanged in the current draft.
The user study presented in this paper does make an effort to address this second issue, by having two different groups of students interact with XploreML either in a lecture or lab setting. However, there is no control group that never works with this tool. Additionally, the survey questions from the user study are about how much one likes the tool (which certainly has its place), instead of an evaluation tool that attempts to determine if working with XploreML leads to deeper understanding of classification.

Pros -
* Interesting and accessible tool (via `shiny`)
* Demonstrates that XploreML can be used in lecture or lab settings

Cons -
* Does not explicitly connect XploreML to learning outcomes
* Difficult to determine the impact of this tool on learning

Minor comments:
* It would be great if the abstract discussed the level of the students that this tool is intended for
* Typo in left side of line 041 "aswell' --> "as well"
* Right side of lines 051 and 052, the authors might consider using a different typeface for the various packages (such as \texttt{})

---

### Decision · Program_Chairs · 2020-07-31

**Decision:**

Reject

**Comment:**

It's been a hard but constructive discussion, but the reviews show from different aspects that this contribution will not be accepted for our workshop. The overall decision was very close.

We encourage the author to keep up their efforts in the field and act upon the suggestions made. We would love to see a submission from you next year. We cordially invite you dial in for the workshop itself to be part of our community and make contributions there. We are looking forward to hearing from you.